# Negotiating Institutional Pathways for Sustaining Climate Change Resilience and Risk Governance in Indonesia

**Jonatan A. Lassa** 

Emergency & Disaster Management, College of Indigenous Futures, Arts & Society, Charles Darwin University, Darwin, NT 0909, Australia; jonatan.lassa@cdu.edu.au; Tel.: +61-8-8946-6756

**Abstract:** Institutions matter because they are instrumental in systematically adapting to global climate change, reducing disaster risks, and building resilience. Without institutionalised action, adapting to climatic change remains ad-hoc. Using exploratory research design and longitudinal observations, this research investigates how urban stakeholders and policy entrepreneurs negotiate institutional architecture and pathways for sustaining climate change adaptation and resilience implementation. This paper introduces hybrid institutionalism as a framework to understand how city administrators, local policy makers, and policy advocates navigate complex institutional landscapes that are characterised by volatility and uncertainties. Grounded in the experience from a recent experiment in Indonesia, this research suggests that institutionalisation of adaptation and resilience agenda involves different forms of institutionalisation and institutionalism through time. Future continuity of adaptation to climate change action depends on the dynamic nature of the institutionalism that leads to uncertainty in mainstreaming risk reduction. However, this research found that pathway-dependency theory emerges as a better predictor for institutionalising climate change adaptation and resilience in Indonesia.

**Keywords:** ACCCRN; Climate change adaptation; institutionalising adaptation; hybrid institutionalism; mainstreaming resilience; urban resilience and adaptation

## 1. Introduction

Many transformative adaptation projects have been exogenously driven by international donors with the aim to build and deepen resilience in both developed and developing worlds. They are often piloted in different ways to build institutional capacity and create institutional pathways that enable local actors worldwide to accelerate urban adaptation [1]. Some of the examples include Asian Cities Climate Change Resilience Network (ACCCRN), 100 Resilient Cities (both funded by Rockefeller Foundation), Making Cities Resilient campaign from United Nations International Strategy for Disaster Reduction (UNISDR), and the UN-Habitat's City Resilience Profiling Programme and many others. These initiatives have been serving as platforms to trigger local adaptation outcomes and disaster resilience.

Responding to the rise of climate risks and disaster vulnerabilities in Asian cities and the needs to build adaptive capacity of the city's governments in Asia, the Rockefeller Foundation, through the $59 million multiyear project, namely ACCCRN, has been supporting 50 secondary cities during 2009–2016 including its two pioneering cities in Indonesia—Semarang City and Bandar Lampung City—to help these cities develop a resilience strategy and build resilience [2].

Three specific adaptation outcomes of ACCCRN in Indonesia include (1) an improved capacity to plan, finance, coordinate, and implement climate change resilience strategies in the selected cities, (2)

shared practical adaptation knowledge to address climate change and deepen the quality of awareness, engagement, demand, and implementation by the selected cities, and finally (3) expansion and/or replication of the ACCCRN models for urban resilience-building in other cities [3,4].

Institutions matter because they are instrumental in systematically adapting to global climate change, reducing disaster risks and building resilience [1]. Without institutionalised action, risk reduction and climatic change adaptation remains ad-hoc in many urban settings. This study investigates the experiments of institutionalising climate adaptation and resilience agendas initiated and implemented by ACCCRN project during 2009–2016 in Semarang City (Figure 1), Indonesia. The key questions include: how urban stakeholders and policy makers negotiate and come to terms with potential forms of institutional scenarios crafted to tackle climate change impacts and disaster risk reduction (DRR) in cities? And how urban stakeholders negotiate institutional pathways for sustaining climate risk governance and achieving resilience? This study contributes to the debate on how to make climate change resilience a local reality by understanding challenges and opportunities faced by local actors in mainstreaming urban adaptation.

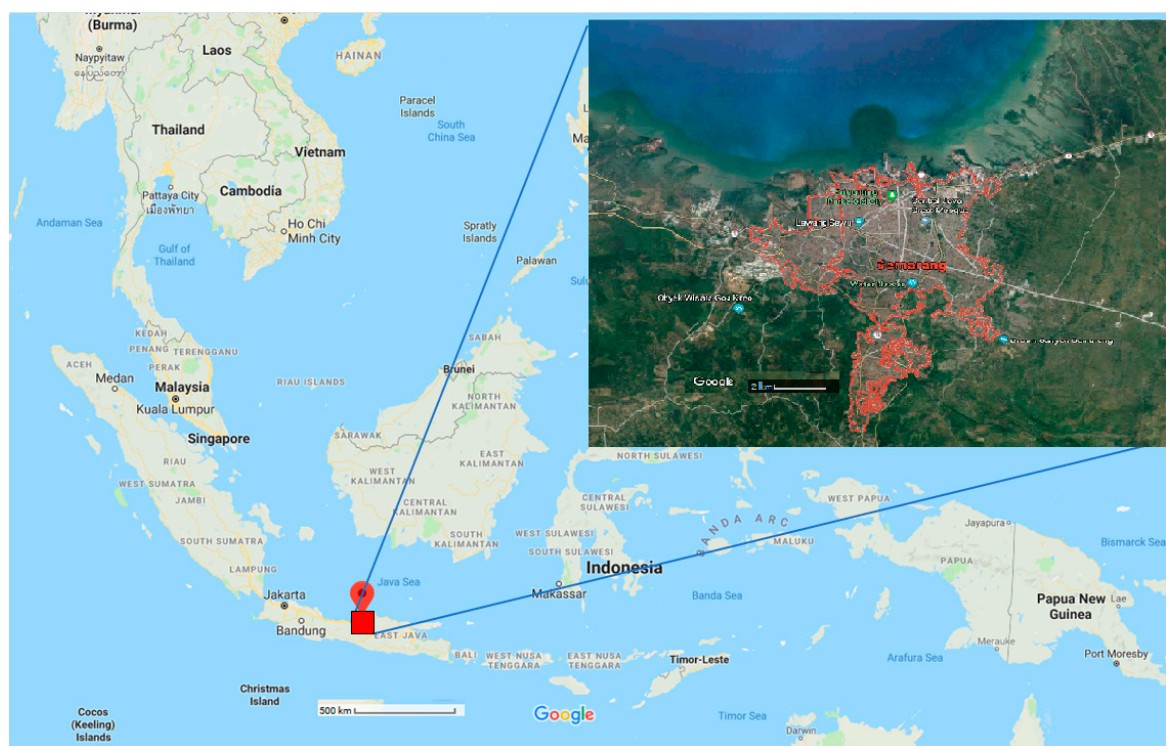

**Figure 1.** City of Semarang Map.

As an institutionalist scholar, the author is motivated to show how institutionalism is used to institutionalise various agendas into institutionalised anticipatory adaptation and disaster risk reduction. This paper adopts the understanding of institutional change as a result of complex interplay between institutions and agents [4]. Institutions refer to formal and informal constraints while agents refer to actors such as local champions and organisations such as local governmental departments and NGOs.

## 2. Theoretical frameworks: Institutions, Institutionalism, and Institutionalisation

Institutions matter because they have become instrumental in making life and death decisions [5]. Institutions not only define what and who will be at risk from climate impacts, but also amend the way risks are defined, perceived, and acted upon [6]. Douglas North argued that "Institutions are the humanly devised constraints that structure human interaction, which are made of: formal constraints

(i.e., rules, laws, constitutions), informal constraints (i.e., norms of behaviour, conventions, self-imposed codes of conduct), and their enforcement characteristics" [7]. North's vision of institutions suggests that institutions structure beyond human-to-human interactions as they also shape human-nature interactions. Unlike the view of behaviourists, institutionalists view institutions as the causality of the communities' behaviour and disaster risk outcomes [8]. Unmanageable risks and occurrence of preventable disasters indicates a lack of political commitments or an absence of public institutions [9]. Interestingly, while many have been working on institutionalisations of and/or mainstreaming climate change adaptation (CCA) and/or disaster risk reduction (DRR) [10–12], the author argues that there is still lack of critical discussion concerning institutions, institutionalism, and institutionalisation as both output/outcomes and process of adaptation and resilience experiments and interventions. In this paper, adaptation is understood as the human systems' adjustments and intervention to 'moderate or avoid harm or exploit beneficial opportunities' [1]. Despite not being synonymous, this paper uses the CCA/DRR as interchangeable as both have shared spaced in reducing risk, adapting to climatic extremes, and building resilience.

Institutionalism is a general approach to understanding institutions [13]. They matter because each type of institutionalism provides the lenses through which resilience initiatives and solutions are institutionalised. The author argues that institutions can be seen as outputs or outcomes of a long process of institutionalisations informed by institutionalism. Institutionalism is the rationality behind both institutions and the process of institutionalisation. For example, one legislative product, like a law, an act, or a bill, is a product of the long process of negotiation, debates and cooperation involving a wider range of actors as well as a process. Furthermore, one cannot study institutions and institutionalisation of CCA and DRR without clearly understanding the school of thought behind the change of and the formation of institutions and the institutionalisation processes that occur in the real world.

The theoretical approach to institutionalism is divided into "old" and "new" schools of thought. The old school emphasizes analysis of the formal-legal and administrative arrangements of government and the public sector [13]. Translating North's vision above to the context of DRR and CCA, institutions can be defined as an admixture of formal rules (e.g., climate-related bills, disaster management acts) [12]. However, the real world is too complex to be seen from a particular view of institutionalism. The "new" institutionalism deals with informal norms (including values, traditions, and beliefs). Both approaches deal with enforcement characteristics (e.g., coercive instruments that regulate land-use and building practices) that shape the landscape of adaptation and disaster risk reduction policy and implementation.

Figure 2 offers the general framework of institutionalisation of a development solution. It argues that an institutionalised action starts from the vision of resilience with a new discourse exercise regarding the change of status quo and the need for resilience and risk reduction. The translation of a resilience vision into institutionalised action depends very much on the types of institutionalism (which will be discuss in Sections 2.1 and 2.2) that will inform the process of mainstreaming and/or institutionalising adaptation and resilience.

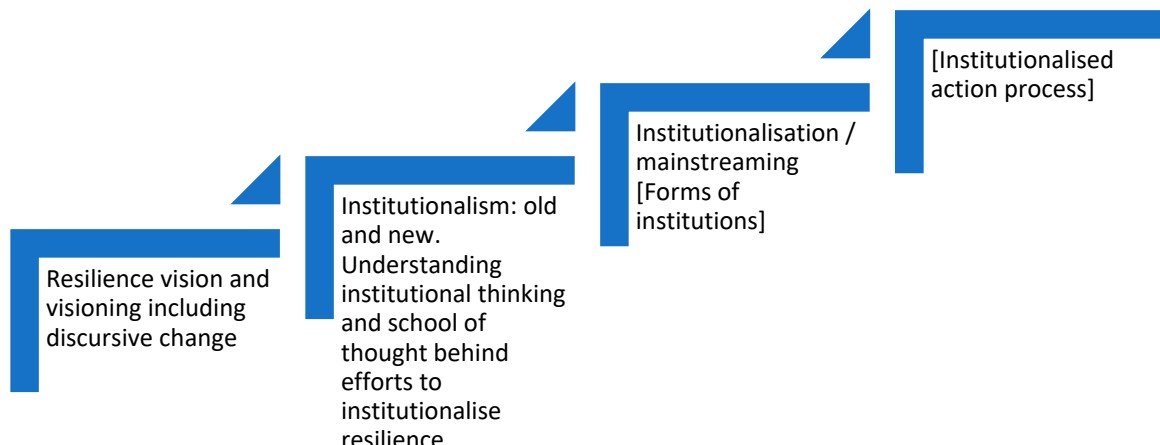

**Figure 2.** Resilience institutionalisation processes. Source: Author, 2019.

*2.1. Old Institutionalism*

A formal approach to resilience has been the main research agenda. For example, researchers have argued that the best instrument for addressing CCA and DRR is to work through the existing formal development process and mechanism [12] which can be defined as existing institutional machineries, ranging from formal bureaucratic processes and routines to existing political and social economic institutions and formal/informal processes that deal with the complexity of urban development. Nevertheless, institutionalisation also includes creating new and amending existing regulations, policies, codes, planning documents, and DRR/CCA-related support programmes [12].

DRR/CCA can be a routine development process as they can be embedded or nested inside existing local mechanism and institutions. Integration of risk and vulnerability information into development planning is an example of routinised adaptation and resilience building [14]. Anguelovski and others [15] defined institutionalisations as "linkage to existing urban planning, decision-making, and governance arrangement" [15].

One of the approaches of old institutionalism is often in favour of the roles of international institutions in shaping disaster and climate policy in developing world via the works of the United Nations and international non-governmental organisations (INGOs). Their initiatives can be seen as exogenous adaptation and resilience which can be transformed into endogenous adaptation through time. However, endogenous adaptation does not negate the need for external actors. Their interactions are structured in a way that it is impossible to understand continuity of DRR and CCA unless—as this paper argues—they are understood as an 'ecosystem of institutionalism' and/or where the real world of CCA/DRR operates according to complex interaction of institutions, institutionalisms, and institutionalisation. In this paper, the word 'institutionalisation' is used interchangeably with 'mainstreaming'.

The Hyogo Framework for Action (HFA) promoted the norm of institutionalisation that is based on a solid legal formal framework such as a specific legislation which eventually enables governments at different levels to develop comprehensive disaster resilience implementation [16]. UN-Habitat also sees the importance of specific climate-related legislation as a legitimate form of institutionalisation [17]. In the context of CCA, this can mean creating a specific administrative task put under existing environmental agencies [12,18]. HFA and Sendai Framework advocate for 'strong basis for implementation' as the institutionalisation process requires a constitutional basis, resource allocation, and the existence of multi-stakeholder platforms to ensure continued commitment and implementation of disaster resilience agenda [19].

## 2.2. New Institutionalism

The new institutionalism is divided into a few categories including the rational choice approach, the historical pathways approach, and the discursive approach [13,20,21].

### 2.2.1. Rational Choice and New Economics Institutionalism

Rational choice institutionalism (ROI) views policy response to climate change as an expression of pure rational choice of local actors to maximise their resilience and safety by adopting adaptation and risk reduction [13]. Regardless of the motivation of international donors, ROI also views governmental institutions and local actors in the developing world as rational agents as they approve any adaptation project based on the interest of their local affairs alone. The typical solution to the adaptation problem is therefore education and capacity building. Unfortunately, there has been mounting evidence that suggests human are not fully rational agents as everyone has limited rationality, and often make foolish decisions due to by-default challenges such as such as imperfect information, lack of motivation, limits of cognitive-ability, time-boundedness, and context [20].

North [7] is one of the key sources for new economic institutionalism (NEI) thinking. NEI views that institutional change from risk-ignorant to risk reduction occur because actors are motivated (or not) by incentives and/or disincentives provided by formal/informal institutions. Therefore, institutions incentivise or disincentivise actors' decisions and preferences to reduce CCA and DRR. This can manifest in the form of projects and resources or the lack of it. Future progress of CCA and DRR is heavily dependent on the institutional context that structures the enforcement of and/or the implementation of CCA + DRR agendas. This theory is often called new institutional economics theory [7] and in this paper, incentive is understood as more than monetary value to also include social, cultural, political, and symbolic incentives [20].

### 2.2.2. Pathway-Dependency Theory

Pathway-dependency often refers to the idea that institutional change occurs not according to rational choice but simply according to historical pathways [20,22]. Local actors' interests in adaptation and resilience is not simply based on knowledge informed by texts books and scientific papers. It is often known as historical regularities in the sense that future pathways (e.g., for urban adaptation and resilience) are simply built on the institutional paths from the past. Institutional pathways also point to the fact that climate disasters and urban crises and their impacts often create complex situations which pose difficulties for local actors to make strategic and rational decisions [22] about CCA and DRR. Consequently, their resilience strategies unfold as they interact with changes in the dynamic relationship between social dynamics and hazardous environments [20,22].

On the contrary, climate change adaptation policy and practice is likely to emerge incrementally in that it involves unpredictable institutional arrangements because ex-ante institutional design might be impossible to be developed by the climate resilience policy entrepreneurs. This implies that resilience strategies at each level of governance are more a result of historical interactions than of anything planned [22]. Such historical interactions include local and international interactions where endogenous climate change policy making can only be made if there is adequate exogenous support that trigger and empower endogenous responses [23].

### 2.2.3. Discursive Institutionalism

Discursive institutionalism aspires institutional changes through the roles of ideas and ideation. The roles of agents' "discursive abilities" [21] is critical to institutional change. This theory assumes a more dynamic and agent-centred approach to institutional change. The institutionalisation of new alternatives or approaches such as climate resilience within an already established institutional stream can be explained by discursive institutionalism. In the real world, roles of local champions can be seen as institutional solutions to unfamiliar agendas like climate change adaptation [24]. Without new ideas

and new ideation exercise, the status quo remains. New ideas provide the opportunity to depart from the status quo. In trying to drive climate change adaptation agenda, discursive abilities of local actors are instrumental for change. The practical instruments for discursive exercises can manifest in the form of local champions [24], public relations and awareness, transmission of knowledge and ideas, training and capacity building, and so on.

*2.3. Hybrid Institutionalism*

Mainstreaming or institutionalising CCA/DRR in a modern urban context requires a multiplicity of efforts. Global champions such as Roberts [10] narrate long-term evolution of the process of climate change institutionalisation in the development context of Durban, South Africa. Roberts offers a practical framework namely 'institutional marker' where she identifies institutionalisation or mainstreaming of resilience via a multipronged approach including: first, the existence of a local champion that serves as a messenger and climate policy entrepreneur, second, the adoption of certain climate change issues in municipal plans, third, resource allocation (human and financial) for climate related issues, and fourth, climate change becomes an important factor in both political and administrative decision making.

Table 1 offers a summary of mainstreaming adaptation options that are complementary in nature. These strategies are flexibly selected by local stakeholders as they see fit and necessary. This suggests that a mixture of formal and informal approaches is needed. Furthermore, a hybrid approach suggests that local reform occurs in the context of complex interaction of local and international actors, as well as state and non-state actors. This also suggests the reality in formal institutional settings are characterised by informalities. This can mean local champions adopt certain ideas or discourse that can be introduced informally. This kind of policy change tends to assume that actors and new ideas must come first (See Roberts [10]). This is later followed by processes (formal and informal) that lead to change in formal development plans and fiscal allocation that occurs in both political and formal administrative settings. Nevertheless, institutionalisation may also mean small changes such as adding new job descriptions of city administrators, training and guidance for local officials, and tweaking developing monitoring and evaluation tools that are sensitive to climate change [1]. Therefore, the author argues that institutionalisation of adaptation and resilience take place at the discretion of and interests of empowered local actors.

*2.4. Climate Risk Governance Concept in Urban Context*

Public governance means governing beyond the conventional governmental power to include different actors including all non-governmental organisation (NGOs) including civil society organisations, non-profit service providers, and business groups [13,20]. Therefore, climate risk governance and/or disaster risk governance concept suggests the polycentric nature of decision making in solving urban climate problem [20]. Furthermore, climate risk governance suggests that there is positive power exercised by external actors as they promote urban climate resilience to be endogenised [23]. ACCCRN is therefore treated here as an example of how urban climate risk governance is exercised where each player ranging from local (from government and NGOs) to international actors (international donors, international NGOs, think tanks) negotiate and shape the process of institutionalising resilience and adaptation.

**Table 1.** Hypothetical options for institutionalisation and mainstreaming adaptation. Source: Author, 2019.

| Types of Institutions | Type of Institutionalisation | Type of Institutionalism | Timelines and Remarks |
|---|---|---|---|
| **Formal approach** | Legislation | Old institutionalism | Long-term implication—Potential stable budget allocation; Required political process and deliberative |
| | Mayor regulation | Old institutionalism | Mid-term; Required executive commitments |
| | Mid-term development plan | Old institutionalism | 5-year period; Depending on drafting process |
| | Annual development plan | Old institutionalism | Short-term; Depending on context |
| | Establishment of specific department | Old institutionalism | Mid- to long-term; Depending on legislative mandates |
| | Fiscal allocation | Hybrid approach | Short-term to long-term: required stable political commitments |
| | Adding specific tasks and agendas to departmental plan | Hybrid approach | Short-term based on local actors' discretionary power |
| | Compliance to United Nations framework | Path-dependency and New economic institutionalism (NEI) | Timelines depending on the nature of the framework |
| **Informal approach** | Ideation through local network | Discursive | Short- to long-term; Depending on incentives |
| | Ideation through local champions | Discursive | Short-term; Depending on incentives |
| | Ideation through international networking | Discursive and NEI | Short- to long-term; Depending on incentives |
| | NGOs/CSOs driven initiatives | Path-dependency theory | Short- to long-term; Depending on incentives |
| | NGOnising formal process | Path-dependency theory | Short- to long-term; Depending on incentives |
| | International initiatives and projects | New economic institutionalism | Short- to long-term; Depending on incentives |
| **Hybrid approach** | Education and training | Rational choice theory | Capacity building |
| | Multi-stakeholder forums or platforms | Discursive | Short- to long-term; Depending on incentives to run the platforms |
| | Vulnerability analysis documents | Discursive | Short-term; Updating is needed—depending on incentives |
| | Shared-learning dialogues | Discursive | Project lifetime; Depending on incentives to run the platforms |
| | Resilience strategy document | Discursive | Short-term; Depending on incentives |
| | Systematic documentation | Discursive | Project lifetime; Depending on incentives |
| | Conference and seminars | Discursive | Project lifetime; Depending on incentives |

Noted: NGO is non-governmental organisations; NGO-nising means, the process of institutionalisation of resilience using an NGO-like structure, such as foundations and/or associations.

### 3. Methods

The author used both exploratory research and longitudinal observations to understand the evolution of institutionalisation processes during the year 2009 to 2016 and post 2016. The participant observations and stakeholder interviews using open-ended interviews with 10 key informants recruited through snowball selection from June–December 2012 (face to face in Semarang). Participant-observation in several meetings including one conference in Surabaya in December 2015. Triangulations were made based on numerous approaches, including desk research, online observations from posted links, ACCCRN websites, published formal planning documents, and City of Semarang websites during 2012–2018. Qualitative analysis was applied to the analysis of the findings.

### 4. Findings: Mainstreaming Adaptation in Semarang City

Unless otherwise state, Section 4 is mainly informed by the field works during 2012 and personal observations including online observation completed during 2015–2018. To avoid confusion, all the personal interviews will be indicated as 'personal communication' when cited in the next sections.

*4.1. Vulnerability and Risk Context*

Semarang City has been increasingly affected by flood risks and severe coastal inundation. There are currently approximately 1.6 million people living in the city with average density of 4000 people/KM-2 in 2016, where some sub-districts, especially its vulnerable North Coast, host more than 11,000 people/KM-2. The city is also sinking due to the high rate of land subsidence which continues to occur at the rate from 1 mm/year to 10 cm/year. Some parts of the North Coast of the city already experience about 12–18 mm/year [25]. As of today, it is estimated that about 7 percent of the city's area have been inundated, significantly affecting a large population and many strategic assets such as the seaport infrastructure [26]. The main reason for the rise in risk level and vulnerabilities is the wide spread of urban settlements that has taken place over the last four decades [25]. The situation is likely to exacerbate in the future due to increases in mean sea level.

*4.2. Asian Cities Climate Change Resilience Network (ACCCRN) Processes and Semarang City Team Formation*

The ACCCRN project seeks to imprint new adaptation pathways within cities through the urban climate governance processes [27] which unfolded in four phases. The first phase (starting 2009—the introductory phase) involved city selection and early shared learning dialogues (SLDs) where city stakeholders were invited to participate and were able to learn and share climate change and other urban development issues [28]. SLDs is critical part of ACCCRN's urban governance framework as it emphasised on capacity building and shared learning [4].

During the second phase (2009–2011), ACCCRN worked through a multi-stakeholder forum, namely City Team, to complete a vulnerability assessment (VA) on the citywide scale. Following the VA, the City Team, in coordination with ACCCRN Indonesia (Mercy Corps) and the Rockefeller Foundation, conducted pilot projects and sector studies such as rainwater harvesting. The third phase (2011–2013) included the completion of the city resilience strategy (CRS) and the development of concept notes towards prioritised intervention (Figure 3). Physical project implementation occurred in the third phase. The final phase included engaging and influencing process at a national level. This included efforts to expand the approach to fifteen other cities that had shown strong interest in replicating resilience building processes [2].

During the early set-up (during the 2009–2011 period), a Semarang City Team (SCT) was formed, composed of an advisory team and a technical team. The advisory team consists of representatives from the city government, led by the city's executive secretary under the Mayor, and the technical team consists of representatives from municipal government agencies, local universities, and local NGOs. The role of the SCT was crucial, as the members were to monitor, control, organise, conduct studies,

manage projects, and report on all activities, processes, and methodologies applied under ACCCRN. The City Team was mandated to lead, facilitate, and catalyse the development of the city resilience strategy document and to institutionalise the strategy for long-term development. SCT had been the backbone of the climate resilience initiatives and emerged as a collective decision-making body.

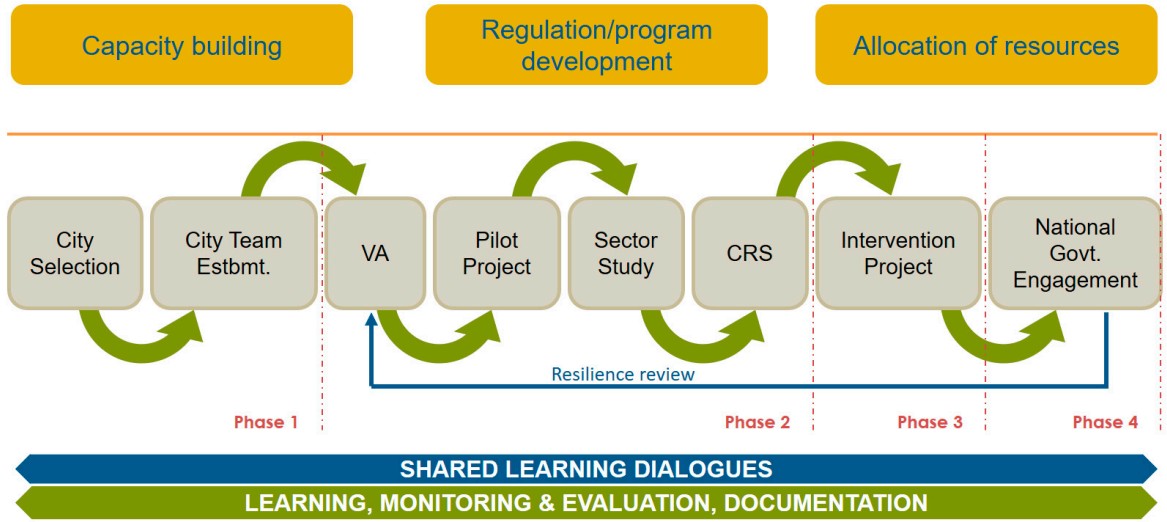

**Figure 3.** Asian Cities Climate Change Resilience Network (ACCCRN) Typical Process in Semarang City. (Source: Sutarto [29])

The ACCCRN Indonesia country strategy 2010–2013 focused on four key activities to ensure continuation of the interventions after the project. First, to manage the project implementation in the city, targeting the capacity-building of City Team members, transforming the City Team into a city climate change resource centre, and facilitating external support for the city government. Physical project implementation included the establishment of flood warning systems and the capacity development of the local communities to conserve the mangrove ecosystem in the coastal areas of the City [29,30]. Second, to disseminate the ideas and educational materials of urban climate change resilience through multiple means such as social networks, conferences, and workshops. Third, to link up with the national government ministries. Fourth, to scale-up projects through the national associations of cities governments and other national and international networks [29].

### 4.3. Kickstart of Project as First Level Institutionalisation

Governments are formal institutions. Therefore, driving new innovation with and by governments requires some formal basis. ACCCRN in Semarang City formally began with decision letters from the mayors. City's leadership was a key variable for this initial stage. The decision letters mandated the formation of the SCT (see Table 2). No other formal regulation was created at the city level to support the initiatives. In Indonesia's legal hierarchy, a decision letter from a mayor does not have good enough power to generate and mobilise both human resources as well as fiscal allocation to implement any innovative action.

**Table 2.** Evolution of climate risk governance in Semarang City. Source: Author, 2019.

| Governance Variables | Agenda 21 | Disaster Management | Urban Climate Change Resilience Arrangement Since 2009 | | |
|---|---|---|---|---|---|
| Timeframe | Semarang Agenda 21 2001 | Existing structure since 2010 | 2009–2011 | 2011–2015 | Since 2015 towards Post-ACCCRN |
| Nature of the organization | Multi-stakeholder forum | Government agency | Multi-stakeholder platform | Multi-stakeholder platform | NGO/association |
| Leadership | Single departmental leader | Single departmental leader | Collective leadership—Coordination team (17 head-of-city departments) | Appointed coordinator—Head of Development Planning Agency | Recruited executives |
| Decision making model | Top down | Command and control—top down | Shared decision making | Shared decision making | Board member |
| Chief executive | Head of Environmental Protection Unit | City Secretary | Head of City Environmental Protection Unit | Head of Planning Division of Development Planning Agency | Executive director |
| Executive secretary | Head of Environmental Protection Unit | Head of BPBD | None | Secretary of Development Planning Agency | NGO Manager |
| Membership | Loose membership | Single agency | 20 members (city government staff, local NGOs and local universities) | 15 members (city government staff, local NGOs and local universities) | Individual members of the present city team |
| Funders | World Bank | APBD | ACCCRN | ACCCRN | External funders |
| Quantity of public consultation | One-off workshop | n/a | Regular meeting—monthly | Regular meeting—monthly | Internal arrangement |

Note: BPBD is the local disaster management office.

Therefore, in order to keep adaptation agenda on top of the city development plan, SCT must be able to create some spaces that allow them to work with limited resources. ACCCRN provided basic resources that can help by jump-starting the city to tackle climate change and urban risks. Two years after the launch of ACCCRN in Semarang, a small reform took place in the government of the City of Semarang where the Department of Development Planning (Bappeda) that used to be a less influential department in the past was revitalised to be a stronger planning institution. Bappeda has been mandated to not only coordinating city planning but also monitoring and evaluating city departments and all sectorial development. In reality, Bappeda had just been functioning as a positive advocate for any innovative policy including climate resilience ([31], personal communication)" Bappeda now is just like a sharp knife' because the agency has started to re-establish its mandates not only as a gatekeeper and quality controller for city development planning agenda but also as a sharpener of development ideas and proposals" ([32], personal communication).

### 4.4. City Resilience Strategy

ACCCRN facilitated the drafting of City Resilience Strategy (CRS)—a fundamental framework that aims at guiding the city to develop policy anticipating and addressing future impacts of climate change. The key features of CRS include a document containing broad adaptation agendas and guidance, prepared by local stakeholders and local government, vulnerability and risk context, organised evidence and analysis justifying adaptation interventions, priorities for resilience actions, consistency with existing planning documents and processes that are fit to local institutional settings, guidance for the private sector and civil society groups to design and implement their own adaptation actions, and linkage and coordination with complementary activities for donors and other funding [3,23,29].

The purpose of the CRS document was also to inform other development policy documents in Semarang City such as the Mid-Term Development Plan (RPJMD) documents. One of the reasons for this adoption is because the Chief Executive of the SCT and the Bappeda Head of Planning Unit happened to be the same person. The City Team was managed under the leadership of city development planning department (Bappeda). This coincidence allowed the CRS to inform the mid-term development plan (RPJMD) for the 2010–2015 period and Semarang Spatial Planning 2011–2030 ([31,33], personal communication). Prior to the CRS document, climate change-related discussion via shared learning dialogues (SLDs) have been directly 'fast tracked' into policy and practices ([34], personal communication). For example, the adoption of CRS recommendations such as rainwater harvesting, and a flood early warning system have been adopted as both policy and programmes in the city [30] and reference ([31,32], personal communication). The inclusion of adaptation agenda depends on the existence and commitments of local champions in the planning process as well as fiscal capability. One of the current issues is, VA and CRS documents might have been outdated and the question is how the actors allocate resources to update the documents?

### 4.5. Local Champion and Leadership

Effective progress towards building resilience in Semarang City has been associated with the roles of local champions. But local champions cannot be easily hand-picked and strategically planned. It took two years for the project to 'recruit' one of the most notable champions, Mr. Purnomo Dwi Sasongko (hereafter Purnomo), who was later elected as the Executive Secretary of the International Council for Local Environmental Initiatives (ICLEI) of Southeast Asia. He joined the SCT in Semarang in 2011 and was soon after promoted as the Head of the Planning and Infrastructure Unit of Bappeda. "Mr Purnomo was the key climate risk communicator and 'ACCCRN spokesperson' to the city's high officials, such as the head of Bappeda and the mayor" ([34], personal communication). He was aware that climate risk communication and strategic sense-making among the city departments are the necessary conditions to create critical awareness regarding the importance of climate integration into city development. Purnomo believed that Bappeda remains the city coordinating body overseeing

annual development plans from 32 departments (a.k.a. local government units shorted as SKPDs) ([31], personal communication).

"Communication is key to all the cities departments. I have to find a proper message that is suitable for the respective department" ([31], personal communication). Purnomo proactively communicates CRS to cities' department focal points by making sense of the need to address climate change within the city departments (SKPDs) ([31,32], personal communication). "To the Public Works department, I could clearly articulate the linkage of drainage maintenance with climate adaptation. As a planner who only recently transformed myself from being a relative climate ignorant to be a climate advocate within the city planning departments, I believe that adaptation can be linked to many urban sectors, from public works to marine and fisheries, water resource management, disaster risk management, environmental services and protection, health, etc. Furthermore, Adaptation is not an extra task for city departments if the key staff in the departments understand how to integrate climate change adaptation into existing issues" ([31], personal communication). Both CRS and VA documents are considered 'academic inputs' to the local government. 'The city needs to deepen the climate adaptation details' ([31], personal communication). Regardless, both CRS and VA were noted as legitimate document that can inform mid- to long-term city planning ([31,34], personal communication).

### 4.6. Resilience Agendas in Development Planning Documents

There is solid evidence that the CRS document has been adopted into the mid-term development plan (RPJMD) 2010–2015, where it mentioned climate change six times including the fact that it recognised climate change impacts to city infrastructure such as road construction, as the rainfall drops become less unpredictable, road construction quality is compromised [35]. Furthermore, the document explicitly shows that climate change becomes a routine business of the Environmental Protection Department and formally budgeted about US$ 1 million a year during 2011–2015 with a focus on urban waste management ([35], Appendix 1 p. 13). Furthermore, there is a clear view that spatial planning is key to mitigate and adapt to climate change ([35] p.V-18 and VI-12).

Interestingly, climate change is still mentioned eight times in the RPJMD 2016–2021, mostly in the introduction. The document also briefly contains risk and vulnerability information. There is an earmarked budgeting for CCA (budget line 2.05.28), allocated for mangrove restoration each year about US$ 275,000 or about a 75 percent decline from previous RPJMD 2010–2015. Unfortunately, it is becoming more realistic than the previous mid-term development plan. Also, the government is aiming at creating 28 climate resilience villages (namely Kampung Proklim) by the end of 2021 ([36], p. VIII-14). The RPJMD 2016–2021 was later revised in 2017, where there is a new budget earmarked as "areas that have the capacity for adaptation and mitigation" with a budget plan from about US$ 425,000 in 2017 to about 625,000 in 2021 to cover all areas ([37], p. VI-14).

### 4.7. Fiscal Capability and Allocation

Cities can generate higher (fiscal) capacity and welfare that can assist their urban population to reduce existing risks [38]. In Indonesia, local opportunities for climate adaptation can be created through self-sustaining incentives that directly shape adaptation imperatives. The increase in cities' capacity to reform their local tax and retribution and curb corruption in tax collection, as demonstrated by Bandar Lampung City [18], is key to boosting fiscal capacity so cities can have greater freedom to invest in the sectors that challenge them most.

The issue is how local actors can sustain the CCA/DRR agenda, commitment, and discourse within the city. One transformative dimension of the ACCCRN initiative is that the Semarang City Planning Agency recently acknowledged the fact that almost all the coastal areas in Semarang are owned by private firms, making it particularly difficult to create mechanisms for coastal protection ([31], personal communication). The city faces serious governance challenges and financial burdens in buying back the coastlines. The SCT influenced the city officials to plan to buy back some of the coastal lands for public access ([31,32], personal communication).

The high visibility of urban risks such as the heavily inundated Northern parts of Semarang City has made it easier for the rest of the city departments to consider climate change adaptation in the annual budget allocation. Transformation is seen at the discursive level. There were adoptions of some proposed activities from the CRS document into annual programme and projects. For instance, at the earmarked budget line, there was no single indication that the Environmental Protection Office (BLHD) had made allocation for addressing the impact of climate change during 2005–2010 budgets. While for the 2011–2015 period, the city started to allocate US$10,000–12,500 annually to either host regular meetings of the SCT or to allocate annual budgets to scale-up rainwater harvesting and mangrove ecosystem development during 2011–2013. Climate change was budgeted under the 5th programme of BLHD, namely 'protection and conservation of natural resources' in 2011–2013. The amount seems trivial, but the discourse behind the allocation is an important first step towards bolder action.

Since the fiscal capacity of many cities in Indonesia has increased in the last 10 years [18], Semarang City has the capacity to fund its own climate adaptation activities in the future. During the 2012 fiscal period, under the flagship of 'climate adaptation', the city managed to allocate US$ 100,000 to buy back some hectares of coastal land ([31–33], personal communication). In the future, the plan to buy back land is expected to continue as long as the land value goes down as a result of nearly permanent inundation in the northern part of the city.

### 4.8. NGOnisation of Formal Platforms?

NGOnisation is defined by the author as a form of institutionalisation in that the multi-stakeholder platform comprises of governmental and non-governmental actors (including NGOs, academics, experts, private sectors and others) have been transformed into into an NGO-like structure. Such transformation, at least in theory, allows the actors to use it as a vehicle to carry on the mandate of adaptation and resilience in more flexible ways.

To ensure that adaptation outcomes can be sustained after the exit of ACCCRN in 2016, the SCT members negotiated to establish a working group or a unit outside the government that may be useful only for conducting objective studies to inform city planning ([31], personal communication). At least three scenarios have been discussed by members of the SCT concerning how to endogenously sustain climate change intervention in Semarang City beyond 2016, after ACCCRN.

The first scenario was a voluntary mechanism. This suggests that committed individuals from city departments who could influence from within using their discretionary roles in planning at sub-department and department levels. Their personal network with the higher authorities was vital to promote the idea of urban sustainability and adaptation. This option was limited. It requires committed leaders at all levels, and reality suggests that the city does not have this luxury all the time ([31,33], personal communication). In fact, there is only a small minority of positive deviant bureaucrats that function as climate goal-keepers in their own departments as well as city secretariat levels.

The second scenario involved mandatory mechanisms, which works through formalised processes with clear mandates and responsibilities as in the case of Surat [39]. Therefore, city resilience-building should be translated into formal rules that provide mandates for the relevant city institutions. This suggests that the city institutions will decide focal points (persons) that may (not) be fit for the task of being climate advocates. This requires two approaches that may complement each other. The first was having a strong formal scenario where there should be a regulation translated into annual operation (e.g., protocol or standard operational procedures) for planning and financing the city. This further requires changes at higher levels, e.g., a Ministry of Home Affairs (MoHA) regulation that mandates city governments to consider climate change as a cross-cutting issue. The second approach is a less formal scenario where local governments, at least at the mayor and legislative levels, can co-create climate regulation/legislation informed by the CRS document.

The later scenario can be achieved through three steps. The first is to create a mayoral regulation that could function for five years, suggesting that city departments work according to the mayor's interests as reflected in the regulation. This can be done at any time as long as the mayor is interested

in and committed to the issue. Second, the mayor's five-year development agenda, reflected in the city's mid-term development plan (RPJMD), must cite or adopt a climate adaptation agenda from the CRS document. This scenario later materialised in the last two planning documents [35–37]. Third, a more long-term approach is that the city can create a local regulation, that can be drafted and endorsed by the mayor with or without the support from the city legislators. The later process involves a lengthy process including political lobbying with some degree of uncertainties.

The third Scenario involves transforming the present SCT into an NGO-like structure ([40], personal communication), where it can maintain its flexibility and interest in promoting climate change adaptation by working through personal contacts among the city's decision-makers. This option, of NGOnisation, in theory could work in the short term. It would already have been exercised in many places in Indonesia where strong environmental NGOs have been working over the last decades. However, strong environmental NGOs do not always succeed in the long run especially when the local political and administrative context change and funding mechanism for NGOs is not certain.

What is interesting is the view of the SCT that functions as a hub of knowledge (or rather as a discursive machine) and technology transfer for climate adaptation. Technology transfer is exemplified by the transfer of technology and knowledge of flood early warning system and appropriate breakwater technologies to protect mangroves [30] and reference ([34], personal communication). Interestingly, the successful technological transfer from ACCCRN can be seen as a direct outcome of its unique approach as it provided multiyear projects that guarantee deeper engagement with local administration and allowing climate change discourse to penetrate in the city structure in a rather informal fashion [30,41].

SCT members are aware of the high turnover of knowledgeable officials at middle and high ranks in public administration in many city/district departments have become a challenge in local governance. This will eventually lead to a lack of institutional memory within many local government offices. New mayors might mean new programmes and new ignorance ([31–34], personal communication). This awareness motivates the empowered officials and stakeholders to develop a structure that allow them to be drivers of CCA knowledge sharing beyond the City of Semarang.

Finally, as implied by all the scenarios above, institutional barriers are not easy to tackle. The SCT finally decided to transform itself into a CSO-like structure namely IUCCE (Initiative for Urban Climate Change and Environment) (Table 1). IUCCE [42] aims to "help achieve the objectives and foster urban resilience to climate change and changing environment". While this problem has been identified, it has become less clear how future climate governance in the city of Semarang will evolve under the UCCE regime.

A closer look at the work of IUCCE indicates that the organisation has been playing roles as a think tank that works beyond the city's jurisdiction. Some of the knowledge products include technical guide on community-based disaster risk reduction, documentation of knowledge concerning mangrove and flood early warning systems etc. While IUCCE can still be functional as an informal "City Team", it is safe to argue that investment of ACCCRN has been clearly successful in terms of knowledge and technological transfer.

### 4.9. 100 Resilience Cities Project in Semarang City

Climate change, including climate risk reduction, remains a marginal task under a few departments including environmental protection department where a few climate adaptation activities (especially mangrove planting) are budgeted for RPJMD 2016–2021 ([36], Section 3.6). While the DRR agenda remains at the discretion of local emergency management agency, for CCA however, based on the lessons during 2011–2015, the actual fiscal allocation does not reflect the budget as proposed by the planners.

After the ACCCRN, the formality of STC is not diminished even though incentives for regular activities, pilot projects, as well as adaptation projects such as flood warning systems ([31], personal communication) have literally come to an end, while at the same time, the Semarang City graduated

from ACCCRN into the new initiatives, namely 100 Resilience Cities project funded by the Rockefeller Foundation [43].

## 5. Discussion

### 5.1. Institutionalisation: Formal and Informal

Old forms of institutionalisation that emphasise the formal approach for reform in formal policy settings remains the imperative of global framework and initiatives. The Hyogo Framework for Action and Sendai's norm of institutionalising resilience can be exemplified by the process of mainstreaming in Surat with the formation of the Climate Change Trust based on the Public Trust Act at state level [39]. However, context shapes the forms of institutionalisation as it can manifest as a general climate policy or specific DRR plan (case of Quito, Ecuador). Institutionalisation can also mean a shift from a resilience strategy that led to the endorsement of the Climate Change Trust (case of Surat, India) [15,39]. Such legal formal achievement in Surat remains to be seen in the City of Semarang as it requires a long local political process. It is also clear that some of the achievements such as resource allocation have been endogenously provided by the executive government in Semarang City, as also noted in the other study such as, Durban [10], Surat [39], and Bandar Lampung City [18,23].

The findings suggest that institutionalisation process take place in several domains, including formal development planning. In terms of institutionalised practice, CCA has become a key task of the local environmental agency while DRR is seen as the task for the local disaster management agency. Such an achievement is predictable, as informed by previous research in different context [12] as well as in Indonesia context [18,23].

In the context of Indonesia's national disaster risk reduction policy reform, institutional change started from legislation and followed by the creation of new administrative units to deal with broader disaster risk problems. Equivalent processes did not occur for climate change adaptation at a national level. In the absence of national guidelines for cities to be adaptive to climate change, secondary cities in the developing world often create their climate adaptation policies and practices through external influences, as exemplified by different ACCCRN cities in India and Indonesia [39,44].

### 5.2. Hybrid Approach to Institutionalisation

Local institutional uncertainty has made future adaptation in cities less predictable. Previous studies suggest that the negative outcomes are particularly due to little stability at public administration and bureaucratic levels, because the local government sector has been affected by dynamic political change and decentralisation. This has been quite clear from the other ACCCRN pioneering cities such as Bandar Lampung [23].

The process of adaptation in cities involved the complex process of exogenous and endogenous efforts in building resilience. While it seems almost impossible to fix the institutional mechanism under the project timeline, the local actors used a rather pragmatic approach to institutionalise climate action. The agenda of hybrid institutionalism has led to pragmatic institutionalisation as it goes beyond what was once seen as multi-pronged approach [10].

Local champions exercised their discursive power. Their network and platform serve as guardians and gate-keepers of city planning, as seen in Western Cape, South Africa, where climate policy entrepreneurs have been the key to adaptation mainstreaming [45]. Their impact can help reduce institutional uncertainty temporarily. The challenge is, local champions and good leadership are often 'given' and cannot be easily planned or recruited in advance.

Responding to the challenges at local and national levels where climate adaptation agenda remains unclear and local capacity remains low, Sharma and Tomar [44] suggested pragmatic solutions namely 'entry points' including embedding adaptation and resilience through existing development and disaster management plans. The first ACCCRN process during 2009–2014 (Section 4.2) are the entry points. To gain quick wins, the 'entry points' approach has also been promoted by some researchers,

such as in Reference [46]. The framing of 'entry points' indicates the nature of exogenous intervention. However, the challenge is how to win at the 'exit point' after the end of international projects remains an important issue for local actors.

The pragmatic approach also requires working with local proponents such as local champions as they can be seen as 'institutions' as they not only create their own rules of the game but play the roles as both goal-keepers and climate policy entrepreneurs. The champions have been the key officials from within existing institutions whose strong passion and interest in promoting innovation within the local government level were seen as vital [47–49] and Reference ([50], personal communication). These champions tend to have a balanced self-interest and public interest in promoting climate resilience agenda not only in Semarang but also in other ACCCRN cities such as Hat Yai City, Thailand [51]. This satisfies the rational choice theory approach as they acted based on their best interest. However, their inability to jumpstart adaptation without external aid suggests that their engagement is largely driven by their interests in incentives created by the projects. This justifies that NEI is the mechanism that helps local actors sustain CCA agendas.

The pragmatic approach to institutionalising CCA also includes the strategy of NGOnisation of the SCT platform. It has been partly used by the actors as a strategy for institutionalising climate change adaptation and resilience building in Semarang City. NGOnisation is an approach where urban stakeholders come to terms with the dynamic nature of complex realities of city development. This mechanism is used to solve institutional uncertainty in the city. While at the same time, the key actors continue to benefit from the existing formal mechanisms, such as continuing to use the platform of STC and others (e.g., 100 Resilient Cities Project funded by the Rockefeller Foundation) and existing departmental commitments related to climatic risks ([50], personal communication). Interestingly, the setup of IUCCE as a think tank in Semarang is favoured by most of the SCT members in Semarang City as they see that this 'NGO-like' structure can provide a balanced self-interest and public interest in promoting climate resilience.

### 5.3. Transforming Urban Adaptation Platforms into Permanent Institution?

Global disaster risk frameworks such as the Hyogo Framework for Action and its predecessor, the Sendai Framework for disaster reduction (SFDRR), have promoted the idea of multi-stakeholder engagement, namely DRR platforms (UNISDR 2005), that are supposed to exist at different levels from global to national to local levels of governance. ACCCRN's City Team in Semarang City can be seen as a CCA/DRR platform or forum. Forums can be seen as institutions as each forum has its own rules of the game. Small groups in any settings that meet regularly suggests that they are bounded by certain values and interests and their rules of the games function as the institutional avenue for them to continue to repeat their interactions [52] until they breakup and the game does not benefit the members.

Therefore, urban adaptation platforms including disaster management platforms are in themselves part of institutional development. Unfortunately, there are always costs associated with regular run of forums and platforms. Empirically speaking, local and national multi-stakeholder platforms have been established in many parts of the world including Indonesia [53], where some were more functional while others were simply function as institutional decoration.

### 5.4. Path-Dependency Theory as a Predictor for Urban Adaptation?

The challenge in the City of Semarang today is not entirely new as in the case of the Semarang Agenda 21 way back in 1997/1998. Semarang City was among the first Indonesian cities to adopt the sustainability framework of Local Agenda 21. The agenda was locally branded as 'Semarang Environmental Agenda: Toward a sustainable city 1998–2003' (hereinafter SEA21) [54]. Semarang City was selected for the pilot project initiated by the World Bank with the acceptance from the Semarang City government ([55], personal communication). Agenda 21 focused on process and trust building as the project conducted extensive consultation with sectoral experts, government officials, NGOs, academics,

and others, which resulted in the 18 chapters of Agenda 21–Indonesia in 1997 [54]. The document identified high- and medium-priority programmes, to be completed in five years (1998–2003) and 10 years (1998–2008), respectively. These priorities included population management, self-resilient community, public transportation, coastal inundation, domestic waste reduction, treatment of human waste, waste management, clean production, healthy rivers, and clean air programmes.

In retrospect, the present institutional pathways for adaptation follow the historical path of the urban sustainability agendas stipulated by SEA21 developed 20 years ago in the same city. To the stakeholders, the most successful contribution of SEA21 to the city administration is the capacity-building ([55], personal communication). The knowledge transfers from the initiative facilitated new awareness for trained staff concerning environmental and urban sustainability. The programme may have been short-lived, but there was a discursive turn within the city's private sectors regarding environmental quality where Bapedalda (Local Environmental Protection Agency or now BLHD) was able to establish a stick-and-carrot approach [56] and reference ([55], personal communication).

Lessons from SEA21 suggested that the process of institutionalising international initiatives such as ACCCRN and others (through formal planning documentation, policy adoption, etc.) often hit the hard wall of local institutions. The issue is not that there was no innovation but innovation in ideas, policies, and practices are often short-lived because such initiatives relied more on persons than systems ([55,57], personal communication). Interestingly, the reliance on persons can be credited as a good start if the persons can play roles as champions with the capability to create adaptation discourse at different levels of governance in cities. The problem is, champions are also timebound. Champions today might not be champions tomorrow as external and internal incentives change through time.

## 6. Conclusions

This research investigates how local actors negotiate to ensure continuity of CCA and DRR as routine development agenda. The question is how urban stakeholders and policy entrepreneurs negotiate and come to terms with potential forms of institutional scenarios crafted to promote adaptation and deepen resilience in the City of Semarang. The whole arrangements and interactions of ACCCRN from the beginning have been about using different forms of institutionalisation to sustain adaptation and resilience agenda. The continuity of CCA and resilience-building depend on mechanisms where there is regular reproduction of resilience discourse including their urgency and importance at different levels and domains ranging from policy documents to the existence of CCA advocates or champions within the agencies in cities. This goes beyond the binary framework of endogenous versus exogenous initiatives for adaptation.

Hybrid institutionalism has merits to provide better understanding of the complexity around institutionalising urban adaptation in Semarang City. While projects such as ACCCRN have co-facilitated processes that aimed at promoting a more rational choice approach to establish a more permanent mechanism, it turned out that such adaptation initiatives have been trapped in the past institutional trajectory such as NGOnising the City Team structure. Lessons from the case of SA21 and the recent development of IUCCE suggests the model of institutionalisation as explained by the theory of historical institutionalism where the 'adaptation pathways' is skewed towards future institutional uncertainty, which makes it difficult for local actors such as the in the City of Semarang to make strategic decisions from within formal institutions [20]. As a result, the STC has transformed itself into an NGO-like structure and served as a think tank instead of policy makers. On the other hand, the emergence of new international initiatives such as 100 Resilient City provides new avenues for the local stakeholders to either restart again or to move forward to the next stages of a city resilient development strategy.

The Semarang City Team has a vision to drive and facilitate a permanent agenda for adaptation and resilience via formal mechanisms. Unfortunately, local dynamic process led these efforts to push the actors from shifting from formal into a more informal approach. The good news is that new

exogenous initiatives remains available via different trajectories as exemplified by the shifts from Semarang Agenda 21, to ACCCRN and to 100 Resilience City programmes. And in between, there is often international frameworks (e.g., among others, the Hyogo Framework or the Sendai Framework) that can be used to ensure resilience discourse remains in the orbits of urban governance.

The author argues that that the (dis)continuity of urban sustainability initiatives, including climate change adaptation and resilience in Semarang city, do occur in the form of hybrid institutionalism but pathway-dependency theory emerges as the most dominant predictor as exemplified by the boom and burst of local platforms ranging from SEA21 to ACCCRN to 100 Resilience City and more into the future. Rational choice thinking and the effort to localise resilience and adaptation often ended up in path-dependency phenomenon where history repeats itself in the form of NGOnising the resilience platforms.

**Funding:** Initial funding was provided by Rockefeller Foundation via ACCCRN Indonesia project managed by Mercy Corps during 2012/2013. Since 2014 this study is self-funded.

**Acknowledgments:** Personal thanks to Ratri Sutarto, Aniessa Delima Sari, Paul Jeffery, Ninik Mulyawati, Omar Saracho (Mercy Crops) who have been supporting all the logistics of this research in Semarang in 2012. Thanks also to Purnomo (The Head of Planning Unit in Bappeda Semarang), Feri Prihantoro, Lilin Budiarti, Lutfi Muhamad, Gunawan Wicaksono, Raharjo Tjahyono and all colleagues at City Team of Semarang and Jawoto Sih Setyono (Diponegoro University) for kindly supporting this research. The author would like to thank the Indonesia Project at Australian National University for the opportunity to present this draft in late 2016. An earlier draft of this article was published as a Working Paper at the Resilience Development Institute. The author would like to thank the three anonymous reviewers who gave very valuable inputs to improve the previous draft. All the mistakes interpreting the information for the sources are the authors.

**Conflicts of Interest:** The author received research funds from ACCCRN Indonesia via Mercy Corps. This study is however independent and critical to satisfy scientific interest alone.

## Abbreviations

| | |
|---|---|
| ACCCRN | Asian Cities Climate Change Network |
| Bapedalda | Local Environmental Protection Agency (old) |
| Bappeda | Local development planning agency |
| BLHD | Local Environmental Protection Agency (new) |
| CCA | Climate change adaptation |
| HFA | Hyogo Framework for Action |
| IUCCE | Initiative for Urban Climate Change and Environment |
| SFDRR | Sendai Framework for disaster risk reduction |
| STC | Semarang City Team |
| UN | United Nations |
| UNISDR | United Nations International Strategy for Disaster Reduction |

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
