# Peer review of "Negotiating Institutional Pathways for Sustaining Climate Change Resilience and Risk Governance in Indonesia"

_climate, doi:10.3390/cli7080095_

Round 1
Reviewer 1 Report
I found the paper is well written in context of institutional pathways for sustaining climate change resilience and risk governance in case of Indonesia. However, I have few minor comments.
1) Methods: I think you have mentioned about the methods used in the study in the last para of Introduction (line 53 -64). It might make more sense if you prepare a separate section of methodology supported by schematic diagram of methodological framework, so that readers will have clear picture and ideas on the methods and they can even apply this framework in their similar studies as well.
2) Study Area: I noticed brief information on the study area in subsection 3.7. Again, it would be great if you could have separate section describing the study area along with the map of study area.
3) While going through the paper, It seems like the term adaptation and resilience have been used synonymously. I think you should clearly define the terms and used in an appropriate ways, so the readers won't get confused.
4) The paper would benefit if you could have elaborated more on risk governance part as well.
5) After having all descriptions from introduction to discussion parts, what is the recommended institutional pathways in the study area to promote CCA and resilience in particular case of your study area, that too supported by your supporting logic/arguments.
Author Response
My responses are provided below each comment:1) Methods: I think you have mentioned about the methods used in the study in the last para of Introduction (line 53 -64). It might make more sense if you prepare a separate section of methodology supported by schematic diagram of methodological framework, so that readers will have clear picture and ideas on the methods and they can even apply this framework in their similar studies as well.
- A new section is created - Section 3. I decided not to add a new figure as it the description has been revised and kept brief.
2) Study Area: I noticed brief information on the study area in subsection 3.7. Again, it would be great if you could have separate section describing the study area along with the map of study area.
- The study area information is now put in Section 4.1 and 4.2. Also, a new figure (2) is added.
3) While going through the paper, It seems like the term adaptation and resilience have been used synonymously. I think you should clearly define the terms and used in an appropriate ways, so the readers won't get confused.
- A brief explanation has been made in Line 73-78.
4) The paper would benefit if you could have elaborated more on risk governance part as well.
- Section 2.4 is created for this (Line 230 - 240)
5) After having all descriptions from introduction to discussion parts, what is the recommended institutional pathways in the study area to promote CCA and resilience in particular case of your study area, that too supported by your supporting logic/arguments.
- Rather than recommending a particular pathways, what I propose is regardless who initiated the process, without understanding the historical pathways, there is a risk of repeating the mistakes from the past. I propose hybrid approach to institutionalising adaptation as the temporary solution (Last paragraph)
Reviewer 2 Report
Manuscript ID: climate-556931
Type of manuscript: Article
Title: Negotiating institutional pathways for sustaining climate change resilience and risk governance in Indonesia
General comments:
This is an interesting article which emphasis on the institutional pathway to sustain urban resilience programs. It is well written. However, I think some improvements are required and for this, I recommend minor revisions.
1. Methodology: How the respondents were recruited? What were the criteria to select the respondents? How many interviews were conducted? What kind of qualitative analysis methods were used? Why a qualitative approach was an appropriate choice for this study?
Some clarifications are required on the methodology of the study. For example, in ‘line 56’, the author mention interviews were conducted between June 2012 to December 2015. However, all the interviews were conducted in 2012 (lines 745-755). Since interviews were the primary data for the study, I prefer to use respondents code number instead of using them as references. Moreover, to ensure the respondents’ confidentiality, I recommend de-identifying their names.
2. The interviews were conducted in 2012. It seems very old data and very difficult to validate the findings with the changing political conditions. The author needs to clarify what measures were taken to validate the reliability of the findings of the study.
3. As a reader, we want to know some issues: Who are you? What is your interest in this topic? What is your investment in this project? What are your intentions? This will help readers to have a fair understanding of the researcher to make the fullest evaluation of the study and to have greater confidence in what they are about to read.
4. In a qualitative study, the readers expect to see some exemplary evidence from the findings instead of just paraphrasing original data. To address this concern, please include a direct excerpt from the respondents’ to support findings.
Technical errors:
1. Spelling mistake: Line 23: (nstitutionalising adaptation)
2. Line 36: The abbreviation is only written for the first time. Later states or the full name or abbreviation, but not both. Same style should be followed in the whole manuscript.
3. Line 77: Elaboration of the term ‘CCA’
4. I recommend using ‘Urban’ as one of the ‘Keywords’
5. References should be organized in ascending order in the whole manuscript.
Author Response
My responses are provided below each comment:
General comments:
This is an interesting article which emphasis on the institutional pathway to sustain urban resilience programs. It is well written. However, I think some improvements are required and for this, I recommend minor revisions.
1. Methodology: How the respondents were recruited? What were the criteria to select the respondents? How many interviews were conducted? What kind of qualitative analysis methods were used? Why a qualitative approach was an appropriate choice for this study?
Some clarifications are required on the methodology of the study. For example, in ‘line 56’, the author mention interviews were conducted between June 2012 to December 2015. However, all the interviews were conducted in 2012 (lines 745-755). Since interviews were the primary data for the study, I prefer to use respondents code number instead of using them as references. Moreover, to ensure the respondents’ confidentiality, I recommend de-identifying their names.
Response: Thank you for this. It is now revised and put under a separate section (Section 3)
2. The interviews were conducted in 2012. It seems very old data and very difficult to validate the findings with the changing political conditions. The author needs to clarify what measures were taken to validate the reliability of the findings of the study.
Response: A legitimate comment. It is now put in Section 3 to make it clearer to readers.
3. As a reader, we want to know some issues: Who are you? What is your interest in this topic? What is your investment in this project? What are your intentions? This will help readers to have a fair understanding of the researcher to make the fullest evaluation of the study and to have greater confidence in what they are about to read.
Response: I am trying to cover this in motivation. But I put it in a way that does not violate the blind review? [Not sure if this is a double blind review]. Thanks
4. In a qualitative study, the readers expect to see some exemplary evidence from the findings instead of just paraphrasing original data. To address this concern, please include a direct excerpt from the respondents’ to support findings.
Noted. I make it clear in Lines 328-330 and Lines 364-366
Technical errors:
1. Spelling mistake: Line 23: (nstitutionalising adaptation) - Correction done
2. Line 36: The abbreviation is only written for the first time. Later states or the full name or abbreviation, but not both. Same style should be followed in the whole manuscript. [Thank you. Done]
3. Line 77: Elaboration of the term ‘CCA’ [Done] changes made from Line 75-80)
4. I recommend using ‘Urban’ as one of the ‘Keywords’ [urban resilience and adaptation is added]
5. References should be organized in ascending order in the whole manuscript [Noted but the template given by the journal is based on Numbered]
Reviewer 3 Report
The authors have made an effort to explaining how climate change adaptation is being mainstreamed into local institutions of Semarang City, by the help of international agencies/policies and in coordination with local administrations. A restructuring of text is required. The paper is too long for a typical paerI have following
1. In abstract, author should begin with a generic statement regarding need of institutions for CCA and DRR, and/or institutions as a catalyst for CCA and DRR.
2. In abstract, “change over time” can be rephrased as temporal change.
3. Typo error in last keyword.
4. Author has often lumped together CCA and DRR together throughout the manuscript, although they are interrelated yet are distinct phenomenon. Author needs to mention relationship between CCA and DRR.
5. P2L60, in my opinion, author can remove description about “structure of paper”, as reader can know it by reading section headings. Rest of para can be merged with previous one, as both points towards methodology of paper. Alternatively, these methodology paras can be moved to section 3.
6. First para of Section 1 represents generic information about institutions. So, P2L67, “Institutions not only… acted upon” can be moved to L75, just before unmanageable risks.
7. Can figure 1 be written as “Resilience [vision and visioning], Institutionalised [action process]?
8. Typo in Figure 1 “Underliying”
9. Last para of Section 2 can be merged last paras of Section 1.
10. Section 2.1 implies “old institutionalism”. In this regard, it’s better to move para 1 of this section and merge it with 3rd para of Section 2.
11. For consistency, mention “Quito, Ecuador”
12. Author has talked about “UNISDR-based” HFA and SFDRR in mainstreaming/institutionalization of CCA. I believe these frameworks only talks about DRR integration. What about IPCC’s agreements/frameworks etc.? Do they mention anything about institutionalizing CCA into governance mechanisms?
13. It would be very beneficial for readers to easily understand theoretical concepts through a framework (figure). Can Figure 1 envisioned into theoretical/conceptual framework? Example Institutionalism in figure 1 can be sub-treed into Old and New, and so on. A little description with each concept can help reader learn a lot about dynamics behind institutions and Institutionalism.
14. Please mention source of table 1 and 2, Author, 2019?
15. P7L232, requires reference.
16. Section 3. is too much detailed. I would advise author to minimize text, and provide information directly related to objectives of the paper. Authors can refer to project reports of ACCCRN, CRS etc.
17. Can discussion section be merged with Section 3 text? It was problematic to connect theoretical concepts of institutionalism with in-situ field practices of urban management.
18. Section 3.9 Was 100 resilient cities is focus of this paper? If not this section can be removed.
19. Section 4.1 Para 1 is redundant, same information is present on P4 122-127. I would advise removing repetition in earlier part of paper.
20. For consistency, move Section 4.3 after 4.4.
21. Lastly, I personally found it difficult to go back and see full names of abbreviations. I would advise author, if possible, to add list of abbreviation at the end of article, which can help readers.
Author Response
My responses are put below each comment in Red:
1. In abstract, author should begin with a generic statement regarding need of institutions for CCA and DRR, and/or institutions as a catalyst for CCA and DRR.
Done. See Line 10-11
2. In abstract, “change over time” can be rephrased as temporal change.
I tried this but then it will change the whole sentence. I put "through time"
3. Typo error in last keyword.
Done [Line 24
4. Author has often lumped together CCA and DRR together throughout the manuscript, although they are interrelated yet are distinct phenomenon. Author needs to mention relationship between CCA and DRR.
Changes made in Line 77-84 to reflect this point
5. P2L60, in my opinion, author can remove description about “structure of paper”, as reader can know it by reading section headings. Rest of para can be merged with previous one, as both points towards methodology of paper. Alternatively, these methodology paras can be moved to section 3.
Done. Deleted.
6. First para of Section 1 represents generic information about institutions. So, P2L67, “Institutions not only… acted upon” can be moved to L75, just before unmanageable risks.
-Done. Thanks - See line 77-79
7. Can figure 1 be written as “Resilience [vision and visioning], Institutionalised [action process]?
8. Typo in Figure 1 “Underliying”
Done
9. Last para of Section 2 can be merged last paras of Section 1.
Done - reflected in line 60-63.
10. Section 2.1 implies “old institutionalism”. In this regard, it’s better to move para 1 of this section and merge it with 3rd para of Section 2.
11. For consistency, mention “Quito, Ecuador”
[Noted]. Done [Line 129]
12. Author has talked about “UNISDR-based” HFA and SFDRR in mainstreaming/institutionalization of CCA. I believe these frameworks only talks about DRR integration. What about IPCC’s agreements/frameworks etc.? Do they mention anything about institutionalizing CCA into governance mechanisms?
[IPCC 2014 does not specify a particular framework but highlight some examples - the table 1 provide more examples than IPCC report]
13. It
would be very beneficial for readers to easily understand theoretical
concepts through a framework (figure). Can Figure 1 envisioned into
theoretical/conceptual framework? Example Institutionalism in figure 1
can be sub-treed into Old and New, and so on. A little description with
each concept can help reader learn a lot about dynamics behind
institutions and Institutionalism.
[I put more wording in the figure to reflect this suggestion. A very good suggestion. Thank you
14. Please mention source of table 1 and 2, Author, 2019?
Done
15. P7L232, requires reference.
This is my statement, so I have changed it into "therefore, the author argues that - See Line 232)
16. Section 3. is too much detailed. I would advise author to minimize text, and provide information directly related to objectives of the paper. Authors can refer to project reports of ACCCRN, CRS etc.
[It is now changed to Section 4. I propose to keep this as it will affect the discussion and conclusion.
17. Can
discussion section be merged with Section 3 text? It was problematic to
connect theoretical concepts of institutionalism with in-situ field
practices of urban management.
Noted. The Table 1 provides a long list of institutinalisation process. This need some demonstration in Section 3. Therefore merging Section 3 and 4 will need a total rework of the structure.
18. Section 3.9 Was 100 resilient cities is focus of this paper? If not this section can be removed.
No. But it is a good example how one intiative ends and new initiative starts in the same city. It is now deleted from the key word. Thanks
19. Section 4.1 Para 1 is redundant, same information is present on P4 122-127. I would advise removing repetition in earlier part of paper.
Many thanks. The Para in P4 is now deleted and moved/integrated with Section 5.1/
20. For consistency, move Section 4.3 after 4.4.
Many thanks. Section 5.4 is now moved to 4.3. Thanks
21. Lastly, I personally found it difficult to go back and see full names of abbreviations. I would advise author, if possible, to add list of abbreviation at the end of article, which can help readers.
Noted. I provide a list of abbreviation at the bottom of the paper. Many thanks